Rare earth element geochemistry of Middle Devonian reefal limestones of the Dianqiangui Basin, South China: implications for nutrient sources and expansion of the reef ecosystem

Mao Qi 1
Gu Shangyi sygu@gzu.edu.cn 1 2
Li Huan 3
Lash Gary G. 4
Zhang Tianyi 1
Xie Xiaofeng 1
Guo Zidong 1
1 College of Resources and Environmental Engineering, Guizhou University , Guiyang , Guizhou , China
2 Key Laboratory of Karst Geological Resources and Environment, Ministry of Education, Guizhou University , Guiyang , Guizhou , China
3 Key Laboratory of Metallogenic Prediction of Nonferrous Metals and Geological Environment Monitoring, Ministry of Education, School of Geosciences and Info-Physics, Central South University , Changsha , Hunan , China
4 Department of Geosciences, State University of New York College at Fredonia , Fredonia , NY , United States of America
Silva Pedro
Electronic publication date: 2022 Jul 22
Publication date: 2022
Volume: 10
Electronic Location ID: e13663
Received 2021 Dec 23; Accepted 2022 Jun 10
Copyright: ©2022 Mao et al.
Copyright year: 2022
Copyright holder: Mao et al.
License: This is an open access article distributed under the terms of the Creative Commons Attribution License, which permits unrestricted use, distribution, reproduction and adaptation in any medium and for any purpose provided that it is properly attributed. For attribution, the original author(s), title, publication source (PeerJ) and either DOI or URL of the article must be cited.
License URL: https://creativecommons.org/licenses/by/4.0/

Keywords: Givetian Period, Turbid water, REEs, Jiwozhai Formation, Nutrients

Funding: National Key R&D Program of China 2018YFC1802601 This work is funded by the National Key R&D Program of China (No. 2018YFC1802601). The funders had no role in study design, data collection and analysis, decision to publish, or preparation of the manuscript.

==============================
The Givetian Age witnessed the greatest expansion of stromatoporoid-coral reefs from low to higher latitudes of the Phanerozoic. Multi-proxy seawater surface temperature reconstruction suggests the establishment of a super-greenhouse climate as a major reason for reef expansion, yet many questions remain. This article presents the results of a rare earth element and yttrium (herein referred to as REY, derived from REE + Y) geochemical study as well as mineralogy and oxygen isotope values of two well-documented Middle Givetian reefal carbonate sections (Jiwozhai and Buzhai) of the Jiwozhai Formation of South China. The nearshore Jiwozhai patch reef succession displays greater biodiversity and more abundant coral than the marginal platform Upper Buzhai reef. Reefal and micritic carbonates of the Jiwozhai section are characterized by shale-like post-Archean Australian Shale (PAAS)-normalized REY patterns, by very weak negative Ce anomaly values (Ce/Ce* 0.80–0.96; average = 0.89), slightly elevated Y/Ho values (28.9–39.1; average = 34.1), and near-unity values of (Pr/Yb)N (average = 0.87), (Pr/Tb)N (average = 0.80), and (Tb/Yb)N (average = 1.09). Moreover, REY patterns of deposits of the Jiwozhai section differ markedly from those of modern seawater. The described geochemical aspects of the Jiwozhai section and the positive correlation of REY and Th contents displayed by the section point to a terrestrial siliciclastic contribution contemporaneous with reef-building. In contrast, REY patterns of the Upper Buzhai reef section samples are similar to those of modern seawater characterized by light rare earth element (LREE) depletion (average (Pr/Yb)N = 0.76), negative Ce anomalies (average Ce/Ce* = 0.88), and average super-chondritic Y/Ho ratios (average = 45.4)). Slightly positive Eu anomalies (Eu/Eu* = 0.93–1.94; average = 1.36) of the Upper Buzhai reef section samples are attributed to the negligible effect of hydrothermal fluids. Middle REE (MREE) enrichment (average (Tb/Yb)N = 1.48) of Buzhai section carbonate samples and positive correlation of REY and Th suggest a riverine input. Combined with siliciclastic mineralogy, oxygen isotope values, and reef-building biota morphology of the studied two sections, we suggest that terrestrial nutrients delivered by rivers far outweighed upwelling as a source of nutrients supplied to the Givetian reef ecosystem of South China. Coral and stromatoporoid in tropic oceans thrived in turbid water containing abundant terrestrial sediment and the nutrient-laden water helped expand reef-builder habitats during the Givetian time.

Introduction

The Devonian Age experienced the greatest expansion of reefs of the Phanerozoic Eon, especially stromatoporoid-coral reefs during Givetian time (Copper, 2001). Reef development during the Devonian extended to latitudes higher than those attained by reefs during the Holocene climatic optimum (Copper, 2001; Jakubowicz et al., 2019). Indeed, the Devonian Laurentian, Russia-Siberia-Kazakhstan, and eastern Gondwana, Sino-Australo-centered reefs have been documented as extending laterally from 400 km to 3,100 km long (Copper & Scotese, 2003). Low latitude Middle Devonian reef-building metazoan communities were dominated by rugose and tabulate corals and stromatoporoids (Copper & Scotese, 2003). However, the cause(s) of global reef expansion during the Middle Devonian time remains ambiguous.

The great expansion of metazoan reefs during the Devonian time was initially attributed to the establishment of super-greenhouse climatic conditions (Berner, 1997; Copper & Scotese, 2003). However, coral bleaching and mortality occur under high temperatures in the modern tropic oceans (Sully & Van Woesik, 2020). Therefore, widespread destruction of reef ecosystems should have been caused in the tropic oceans during the Givetian Age. It is not the case. Climate and water quality affect coral reef growth and reef ecology in modern oceans (Pandolfi, 2015). Regardless of water temperature, nutrients, and sediment abundances of seawater impact modern coral reef systems (Rogers, 1990; McCulloch et al., 2003; Erftemeijer et al., 2012; Zaneveld et al., 2016).

Modern reef-building corals are known to flourish in oligotrophic waters. Moreover, the combination of increasing nutrient levels and rising seawater temperature is known to be responsible for the decline of coral ecology in recent years (Wiedenmann et al., 2013; Hughes, Day & Brodie, 2015; Rädecker et al., 2021). Terrestrial runoff supplies nutrients as well as sediments into the oceans. While coral reef systems are threatened by increasing terrestrial runoff, it is documented that turbid nearshore environments show reduced bleaching of corals under high temperatures (Sully & Van Woesik, 2020). Some workers speculate that changing nutrient types and availabilities in Phanerozoic oceans exerted some degree of control on reef distribution (Wood, 1993; Kiessling, 2001). However, this concept suffered from a lack of robust evidence of nutrient levels (Copper & Scotese, 2003).

Particle reactive rare earth elements and yttrium (REY), nitrate, phosphate, and silica abundances of modern seawater display similar vertical distribution profiles (Byrne & Kim, 1993; Schijft, Christenson & Byrne, 2015). Furthermore, REY, because of the similar ionic radii of REY and Ca2+, is incorporated into inorganic and biogenic carbonate minerals (Swart, 2015). The large distribution coefficient of REY between carbonate minerals (e.g., calcite and aragonite) and seawater make REY concentration and distribution in carbonate rocks resistant to the effects of diagenesis and dolomitization (Banner & Hanson, 1990; Webb et al., 2009; Liu et al., 2019). Consequently, REY is commonly applied to the analysis of trace marine nutrient levels, terrestrial flux, and water mass transport in modern ocean and coral reef ecosystems (Hara et al., 2009; Grenier et al., 2018; Leonard et al., 2019; Pham et al., 2019; Saha et al., 2021). The fact that ancient limestone and modern coral can serve as seawater chemistry proxies (Nothdurft, Webb & Kamber, 2004) validates the use of carbonate REY geochemistry as a means of reconstructing changes in the nutrient type and abundance in carbonate deposits. Pristine rare earth element signatures from carbonates could be retrieved for deep-time environmental reconstruction after careful evaluation (Zhao et al., 2021; Zhao et al., 2022).

The Middle Devonian Givetian reef tract of the Dianqiangui Basin of South China extends from slope to near-shore environments for more than 1,700 km (Wu et al., 2010). The sedimentary succession, details of reef facies, and biodiversity of reef deposits of South China have been thoroughly studied in past decades (e.g., Wang et al., 1979; Wang, 2001; Liu et al., 2004; Huang et al., 2020). The present paper considers REY geochemistry, mineralogy, and oxygen isotopic values of Middle Devonian Givetian reefal carbonates of the Jiwozhai patch reef and Buzhai platform margin reef in Dushan County, South China. This study aims to decipher the sources of nutrients delivered to the Givetian reef complex of South China as well as the role that terrestrial flux played in the maintenance of reefs during the Givetian reef expansion.

Geological Setting

The Dianqiangui Basin of the South China Block was located near the equator in the eastern part of the Palaeo-Tethys Ocean during the Givetian Age (Huang et al., 2020; Fig. 1A). The Palaeo-Tethys, bordering the northern margin of Gondwana, formed ∼400–385 Ma following the Kwangsian Orogeny (Xian et al., 2019; Qiu et al., 2020). Marine transgression was associated with rift-related movement of the South China Block in Early Devonian time (Qie et al., 2019). The paleogeography of South China experienced significant change during the Givetian time in association with syn-depositional rifting. The basic volcanoes erupted in Dachang of Guangxi from the deepwater basin. The basalt in Guangxi is about 30–100 m thick. The mineralogy of the basalt is characterized by porphyritic plagioclase with pyroxenes and minor olivine. The overlying and underlying rocks of the basalt contain a lot of marine fossils such as tentaculites, which are interbedded with cherts and carbonates, and the submarine eruption was invoked as the origin of the basalt (Liu, Qin & Yan, 2012). The Givetian also was the acme of reef development in the Phanerozoic Era during which small multi-cycle reefs formed on the inner platform of South China (Wu et al., 2010).

Figure 1 (A) Givetian paleogeography (modified from Huang et al., 2020) showing the location of the Dianqiangui Basin (DB); (B) palaeogeography of Dianqiangui Basin during the Givetian Stage (modified from Huang et al., 2020); red stars indicate the locations of the studied Buzhai (BZ) and Jiwozhai (JWZ) reef sections; white rectangle shows the location of Fig. C; (C) map of the study area and its location in China; red stars show the locations of the Jiwozhai (JWZ) and Buzhai (BZ) reefs.

A and C: ©Elsevier B.V. https://doi.org/10.1016/j.palaeo.2020.109895.

The two studied reef sections expose the Jiwozhai Formation of Dushan County of the Guizhou Province, China (Figs. 1B and 1C). The Jiwozhai Formation is overlain conformably by silty shale of the Hejiazhai Member of the Upper Devonian Wangchengpo Formation and is in conformable contact with underlying quartz sandstone of the Songjiaqiao Member of the Middle Devonian Dushan Formation (Fig. 2). The brachiopod assemblage Stringocephalus burtini-Undispirifer undiderus and rugose coral assemblage Endophyllum guizhouense-Sunophyllum elegantum confirm a Givetian age of the Jiwozhai Formation (Liu et al., 2004; Qie et al., 2019; Huang et al., 2020).

Figure 2 Stratigraphic columns of the Upper Buzhai and Jiwozhai reef sections.

Source credit: Shang Yi Gu.

Gray to dark-gray medium- to thick-bedded interlayered micritic limestone and marl comprise the primary lithology of the Jiwozhai Formation with scattered reef limestone deposits in the lower part of the unit. Variations in thickness and lithology of the Jiwozhai Formation between the studied sections likely reflect differences in paleogeographic locations of deposition. Patch reefs (e.g., Jiwozhai patch reef) in the nearshore and abundant fringing reefs (e.g., Buzhai reef) on the platform margin comprise the dominant reef types of the Jiwozhai Formation. The enhanced biodiversity displayed by the Jiwozhai reef has been thoroughly described by Huang et al. (2020) and includes laminar stromatoporoids and tabulate corals as the prevailing reef builders. The upper reef deposits of the Jiwozhai Formation (hereafter referred to as the Upper Buzhai reef) and lower reef strata of the overlying Jipao Member of the Dushan Formation comprise the Buzhai reef. The well-documented Buzhai reef is dominated by laminar stromatoporoids.

The Upper Buzhai reef section (GPS 25°50′56.12″N, 107°34′32.74″E) is located in Dongyao village along a country road (Fig. 3A). The reef core facies from the Upper Buzhai is about 3,000 m along the strike, 600 m along with the trend, and about 210 m in thickness (Liu et al., 2004). The reef section is situated in the reef flank facies and comprises three alternating reef limestones and bioclastic packstones; the stratigraphically lowest quartz sandstone separates the Jiwozhai Formation from the underlying Dushan Formation (Fig. 2). The lower reef interval of the Upper Buzhai reef section is about 2.9 m thick and made up largely of laminar stromatoporoids (Fig. 3B) and subordinate sponge (Fig. 3C), and tabulate and rugose corals (Figs. 3D–3E). Brachiopods (e.g., Stringocephalus, Fig. 3F) appear to have been present as reef dwellers. The middle reef interval is about 6.1 m thick and dominated by laminar stromatoporoids and is separated from the lower reef interval by about 2.0 m of bioclastic packstone. The upper part of the studied Jiwozhai Formation succession exposes approximately 8.7 m of reef limestone and 15.1 m of medium- to thick-bedded bioclastic packestone (Fig. 2), which is overlain by silty shale of the Wangchengpo Formation. A detailed biota assemblage study suggested that the Upper Buzhai reef was developed near or below the fair-weather wave base (Liu et al., 2004).

Figure 3 Field photos and photomicrographs taken under polarized-light of Buzhai reef outcrops and samples.

(A) View of the Buzhai reef section; (B) close-up of the reef-builder stromatopora; (C) sponge fossil (center of photo); (D) photomicrograph of tabulate coral; (E) photomicrograph of rugose coral (F) close-up view of Stringocephalus sp. Source credit: Shang Yi Gu.

The Jiwozhai patch reef section (GPS 25°50′56.12″N, 107°34′32.74″E) is located in Dahekou Geopark in Dushan County and comprises three reef and bioclastic wackestone and packstone intervals (Fig. 2). The Jiwozhai patch reef is much smaller in size than the Upper Buzhai reef. The patch reef is about 100 m–120 m wide and 4.5 m–8 m thick. The Upper Buzhai reef section contains more reef limestone than is present in the Jiwozhai section whereas the latter contains more muddy limestone (Fig. 4A) and a greater benthic fauna. Also, laminar stromatoporoids (Figs. 4B–4E), tabulate corals (Figs. 4C and 4F), and chaetetids (Fig. 4G) appear to have been the dominant reef builders at the depositional location of the Jiwozhai section. Rugose corals (Figs. 4B–4D) and brachiopods (Fig. 4H) were also present but were subordinate numbers. Abundant laminar stromatoporoids and platy tabulate corals imply that the Jiwozhao reef was formed in very shallow seawater with a depth below 10 m.

Figure 4 Field photos and photomicrographs taken under polarized-light of Jiwozhai reef section and outcrop samples.

(A) General view of the Jiwozhai patch reef; (B) close-up of rugose coral (Ru) and stromatopora (St); (C) close-up of rugose (Ru) and tabulate (Ta) corals; (D) photomicrograph of rugose coral; (E) photomicrograph of laminar stromatopor; (F) photomicrograph of tabulate coral; (G) Chaetetid (Ch) encrusting tabulate (Ta) coral; (H) photomicrograph of brachiopod.

Materials and methods

We measured the Upper Buzhai reef and Jiwozhai patch reef sections of the Jiwozhai Formation and collected 34 fresh rock samples for mineralogy, lithology, trace element and oxygen isotopic analyses. Veining and weathered carbonate samples were avoided from field outcrops. One part of each specimen was used for the preparation of thin sections for petrographic observation. The other part was micro-drilled from stromatoporoids (reefal limestone) or micrite (wackestone and packstone) with a tungsten carbide bit to obtain powder for mineralogy, trace element, and oxygen isotope analyses, and again the veining and weathered parts were avoided. Samples are numbered in order from bottom to top as BZ-1 to BZ-20 for the Buzhai section and JWZ-1 to JWZ-14 for the Jiwozhai section (Fig. 2). Powders of each sample were analyzed for mineralogy, rare earth, and other trace elements. Carbonate oxygen isotope values are also analyzed. Thirty-four thin sections were produced for visual inspection under the polarized light microscope in the Key Laboratory of Geological Resources and Environments, Guizhou University, Ministry of Education, China.

Trace element analyses were performed at Guizhou Tongwei Analytical Technology Co., Ltd. on a Thermo Fisher iCAP RQ ICP-MS equipped with a Cetac ASX-560 AutoSampler. Approximately 50 mg of each rock powder was dissolved in a Teflon bomb with a double-distilled concentrated HNO3-HF (1:4) mixture. The dissolution was maintained in an oven at 185 °C for 3 days. The solutions were then dried down to evaporate HF. The sample residues were re-dissolved with double distilled concentrated HNO3 followed by 1:1 HNO3 and dried again. Then, the samples were dissolved in a final 3ml 2N HNO3 stock solution. Finally, the sample solution was diluted to 4,000 times with 2 percent HNO3 and added with 6 ppb Rh, In, Re, and Bi internal spikes. USGS standard W-2a was used as reference standard and crossed checked with BHVO-2 and other reference materials. Instrument drift mass bios were corrected with internal spikes and external monitors. The ICP-MS procedure for trace element analysis follows the protocol of Liang, Jing & Gregoire (2000). The analytical error for REE and other trace elements is less than 5%.

Mineralogical analyses were performed by X’Pert Powder XRD analyzer (operating at 40 kV and 40 mA) using Cu Kα radiation in Guizhou Key Laboratory of Comprehensive Utilization of Non-metallic Mineral Resources of Guizhou University. Powder XRD patterns were collected in the 2θ range of 5°–110° with a step size of 0.05°. The reference intensity ratio method is applied to estimate the relative contents of calcite and dolomite in the samples.

Carbonate oxygen isotopic values were performed in the Key Laboratory of Karst Geo-resources and Environment of Guizhou University with Thermo Fisher Delta V Advantage stable isotope mass spectrometry. The brief analytical procedure is as follows. About 100 µg of carbonate powder was introduced into the reaction bottle and then sealed. The reaction bottle was blown with helium for 330 s. Phosphoric acid was then mixed with the carbonate powder to release CO2 at 70 °C for 1 h. Finally, the released CO2 was sent by helium gas from Gasbench to Delta V Advantage for isotope analysis. The isotope data are reported as δ18O relative to VPDB standards. Reference material NBS18 was used for quality control. The analytical precision is ±0.20 ‰.

Results

Measured concentrations of rare earth and other trace elements, oxygen isotopic values, and mineral contents of the Jiwozhai Formation samples are presented in Tables 1 and 2. Post-Archean Australian Shale (PAAS)-normalized element ratios (Pr/Yb)N, (Pr/Tb)N, and (Tb/Yb)N were calculated to define the degree of fractionation between light REE (LREE) and heavy REE (HREE), light REE and middle REE (MREE), and middle REE and heavy REE, respectively. Y/Ho ratios were calculated without normalization. Some rare earth element anomalies were calculated on a linear scale as the following (Lawrence et al., 2006): Ce/Ce*=Ce/(Pr*Pr/Nd)

Eu/Eu*=Eu/Sm2/3∗Tb1/3.

Table 1 Rare earth and other trace element concentrations (mg/kg), and oxygen isotopic values (‰) in samples of Jiwozhai reef section.

	JWZ-1	JWZ-2	JWZ-3	JWZ-4	JWZ-5	JWZ-6	JWZ-7	JWZ-8	JWZ-9	JWZ-10	JWZ-11	JWZ-12	JWZ-13	JWZ-14	
La	26.6	16.5	6.45	10.4	4.96	1.86	12.1	0.80	11.2	6.21	3.46	3.43	0.41	1.06	
Ce	55.8	29.4	12.5	16.9	8.88	2.99	22.7	1.64	19.9	10.4	6.42	6.55	0.77	1.86	
Pr	6.46	3.40	1.59	1.99	1.10	0.38	2.71	0.21	2.31	1.19	0.79	0.93	0.09	0.22	
Nd	23.7	12.3	6.03	7.15	4.16	1.50	10.2	0.87	8.35	4.28	2.96	3.66	0.33	0.84	
Sm	4.33	2.19	1.18	1.35	0.82	0.31	1.93	0.17	1.53	0.79	0.57	0.78	0.07	0.16	
Eu	0.85	0.41	0.26	0.28	0.18	0.07	0.40	0.04	0.32	0.17	0.12	0.16	0.01	0.04	
Gd	3.91	1.91	1.09	1.23	0.76	0.32	1.75	0.19	1.36	0.74	0.52	0.71	0.06	0.16	
Tb	0.65	0.29	0.18	0.20	0.12	0.05	0.28	0.03	0.22	0.12	0.08	0.11	0.01	0.02	
Dy	3.82	1.68	1.02	1.17	0.71	0.32	1.57	0.18	1.29	0.71	0.50	0.64	0.06	0.14	
Ho	0.78	0.36	0.21	0.25	0.16	0.08	0.33	0.04	0.28	0.16	0.10	0.14	0.01	0.03	
Er	2.22	1.05	0.59	0.77	0.44	0.22	0.97	0.10	0.85	0.47	0.31	0.38	0.03	0.09	
Tm	0.35	0.16	0.09	0.12	0.07	0.03	0.15	0.01	0.13	0.07	0.05	0.06	0.00	0.01	
Yb	2.21	1.06	0.56	0.77	0.41	0.21	0.94	0.08	0.88	0.47	0.30	0.36	0.03	0.08	
Lu	0.34	0.17	0.09	0.12	0.06	0.03	0.15	0.01	0.14	0.08	0.04	0.06	0.00	0.01	
Y	23.2	11.5	7.42	8.56	5.72	3.09	10.8	1.49	8.84	5.43	3.59	3.99	0.42	1.06	
Ba	272	194	88.1	208	76.6	30.7	203	33.4	242	115	50.6	58.8	14.1	28.1	
Mn	913	761	343	275	266	104	363	198	215	197	114	154	131	358	
Sr	57.6	63.9	549	478	543	445	421	668	326	331	428	278	187	95.1	
Zr	134	69.6	25.4	46.1	17.3	6.88	58.6	2.41	59.5	29.8	15.2	19.0	0.69	2.68	
Th	12.0	6.97	2.23	4.29	1.86	0.73	4.68	0.19	4.45	2.31	1.17	1.35	0.06	0.29	
δ18Ocarb			−9.99	−10.11	−6.70	−9.94	−9.47	−10.33	−9.78	−7.87	−9.31	−9.23	−18.72	−19.44	
Notes.

Sample JWZ-1 is quartz sandstone from Dushan Formation; sample JWZ-2 is muddy limestone from the bottom of the Jiwozhai Formation; Samples from JWZ-3 to JWZ-9 are reefal limestone and the other samples are bioclastic packstone from Jiwozhai Formation.

Table 2 Rare earth and other trace element concentrations (mg/kg),and oxygen isotopic values (‰) in samples of the Upper Buzhai reef section.

	BZ-1	BZ-2	BZ-3	BZ-4	BZ-5	BZ-6	BZ-7	BZ-8	JBZ-9	BZ-10	BZ-11	BZ-12	BZ-13	BZ-14	BZ-15	BZ-16	BZ-17	BZ-18	BZ-19	BZ-20	
La	14.7	16.5	8.77	2.38	2.82	5.18	7.57	2.54	0.46	2.74	0.46	0.36	2.73	7.68	4.18	1.79	1.21	0.44	12.8	13.2	
Ce	31.2	32.9	16.9	3.38	4.49	9.54	14.6	2.64	0.91	4.11	0.62	0.64	4.01	12.6	6.88	2.44	1.99	0.58	29.8	28.4	
Pr	3.46	3.75	2.16	0.48	0.59	1.40	1.72	0.47	0.14	0.60	0.09	0.09	0.54	1.55	0.92	0.30	0.27	0.09	3.08	3.26	
Nd	13.1	14.4	8.65	1.94	2.43	6.45	6.87	1.94	0.63	2.54	0.41	0.44	2.19	5.79	3.71	1.11	1.10	0.41	11.9	12.8	
Sm	2.55	2.78	1.77	0.44	0.70	1.85	1.49	0.40	0.14	0.65	0.09	0.10	0.47	1.06	0.82	0.21	0.24	0.08	2.46	2.66	
Eu	0.41	0.48	0.37	0.20	0.32	0.459	0.30	0.13	0.05	0.18	0.02	0.03	0.13	0.27	0.21	0.07	0.06	0.03	0.43	0.46	
Gd	1.81	2.36	1.89	0.61	0.98	2.09	1.61	0.50	0.18	0.72	0.10	0.11	0.48	1.10	0.88	0.25	0.26	0.10	1.92	2.18	
Tb	0.24	0.33	0.30	0.10	0.16	0.281	0.27	0.08	0.03	0.11	0.02	0.02	0.07	0.17	0.13	0.04	0.04	0.02	0.26	0.31	
Dy	1.31	1.82	1.74	0.60	0.91	1.43	1.52	0.55	0.17	0.65	0.11	0.09	0.42	0.96	0.78	0.23	0.23	0.10	1.29	1.61	
Ho	0.26	0.36	0.35	0.13	0.19	0.261	0.30	0.14	0.04	0.13	0.02	0.02	0.09	0.20	0.16	0.05	0.05	0.02	0.25	0.31	
Er	0.77	1.03	0.92	0.34	0.48	0.624	0.84	0.39	0.10	0.36	0.07	0.05	0.24	0.56	0.42	0.14	0.13	0.06	0.72	0.88	
Tm	0.13	0.17	0.14	0.05	0.07	0.080	0.12	0.06	0.01	0.05	0.01	0.01	0.04	0.08	0.06	0.02	0.02	0.01	0.12	0.14	
Yb	0.85	1.07	0.78	0.25	0.37	0.425	0.74	0.34	0.06	0.30	0.05	0.04	0.20	0.48	0.36	0.11	0.10	0.05	0.82	0.95	
Lu	0.13	0.17	0.12	0.04	0.05	0.061	0.11	0.05	0.01	0.04	0.01	0.00	0.03	0.07	0.05	0.02	0.02	0.01	0.13	0.14	
Y	7.09	10.9	12.2	6.19	8.13	10.2	9.86	7.18	1.99	6.22	1.29	0.98	4.12	8.17	6.45	2.64	2.09	1.11	6.64	8.26	
Ba	75.5	60.5	55.8	70.4	18.6	28.5	46.7	19.5	5.85	14.9	2.89	15.3	34.2	215	34.1	9.28	8.60	5.60	90.2	96.9	
Mn	62.7	192	137	395	148	149	181	155	122	98.3	78.5	69.0	116	126	105	81.8	62.7	106	32.6	104	
Sr	164	149	259	344	200	286	347	214	117	257	174	139	241	358	296	290	232	177	232	206	
Zr	141	136	53.1	6.83	4.35	13.3	14.3	3.15	0.30	3.54	0.54	0.49	4.10	15.7	7.46	2.33	1.41	0.37	107	96.0	
Th	6.95	6.44	2.91	0.29	0.42	1.37	2.21	0.34	0.02	0.46	0.02	0.02	0.57	2.24	1.16	0.26	0.25	0.06	6.65	6.50	
δ18O			−6.93	−6.77	−6.84	−7.10	−5.05	−4.87	−5.17	−4.98	−5.55	−5.48	−4.25	−5.41	−15.80	−16.12	−8.28	−7.15			
Notes.

BZ-1 and BZ-2 are quartz sandstone samples from Dushan Formation; samples BZ-4, BZ-5, BZ-8, BZ-9, from BZ-11 to BZ-16 are reefal limestone, and the other samples are Bioclastic packsttone from Jiwozhai Formation; BZ-19 and BZ-20 are siltstone samples from Wangchengpo Formation.

Jiwozhai reef section

Carbonates of the Jiwozhai reef section have total REY (TREY) concentrations ranging from 2.31 to 82.38 ppm (average = 32.65 ppm). The average concentration of REY of analyzed reefal limestone samples is 37.28 ppm, greater than the 16.22 ppm average of analyzed bioclastic packstone samples. One quartz sandstone sample (JWZ-1) contains the greatest TREY concentration of 155.2 ppm whereas the muddy dolostone sample (JWZ-2) is characterized by a slightly greater TREY concentration of 82.38 ppm. All analyzed samples regardless of lithology display flat (shale-like) PAAS-normalized REY patterns defined by minor negative Ce anomalies and weak fractionation among LREE, MREE, and HREE (Fig. 5). Ce anomalies of carbonate samples vary from 0.80 to 0.96 (average = 0.89). Eu anomalies for all analyzed samples fall between 0.91 and 1.19 (average = 1.05) (Fig. 5). Average (Pr/Yb)N, (Pr/Tb)N, and (Tb/Yb)N ratios of analyzed carbonate samples are 0.87, 0.80, and 1.09, respectively, are similar to those of the siliciclastic rock sample (0.93, 0.87, and 1.07, respectively). Carbonate samples are characterized by Y/Ho ratios of 28.9 to 39.1 (average = 34.1), slightly greater than the 29.9 Y/Ho value of the siliciclastic rock sample. However, one reefal limestone sample with high carbonate contents (JWZ-6) captured oceanic signals characterized by the lowest (Pr/Yb)N ratio (0.60) and the highest Y/Ho ratio (39). Mn/Sr values of analyzed limestone samples range from 0.23 to 0.86 (average = 0.53), and one dolostone sample (JWZ-14) has the highest value of 3.76.

Figure 5 PAAS-normalized REY patterns for Jiwozhai reef section samples.

Modern seawater and river water REY patterns are from Wang et al. (2018). Refer to Fig. 2 for sample locations.

Contents of Th and Zr of the analyzed siltstone sample are 12 ppm and 134 ppm, respectively. Thorium and Zr concentrations of carbonate samples are one to three orders of magnitude less than those of the quartz sandstone sample and display a significant positive correlation (r2 = 0.964, N = 13; Fig. 6A). TREY and Th also display a strong positive co-variance (Fig. 6B). Thorium and detrital minerals (quartz and illite) display a strong positive correlation (r2 = 0.961, n = 14). These results are consistent with the mineralogy of the samples with the higher content of quartz and illite corresponding to the higher level of TREY. Both the reefal limestone and packstone samples have varied content of detrital minerals (quartz and illite) with quartz being predominant in the Jiwozhai section (Table 1).

Figure 6 (A) Th vs. Zr cross-plot for Jiwozhai reef section samples; (B) Th vs. TREY cross-plot for Jiwozhai section samples.

The δ18Ocarb values for the Jiwozhai reef section show a decreased trend from −9.99 ‰ to −19.14 ‰ along with the stratigraphic height. The top two samples (JWZ-13 and JWZ-14) of the section have the lowest δ18Ocarb values of −18.72 ‰ and −19.44 ‰, respectively.

The Upper Buzhai reef section

PAAS-normalized REY patterns for the Upper Buzhai reef section sample suite are presented in Fig. 7. Four samples of quartz sandstone and shale (BZ-1, BZ-2, BZ-19, and BZ-20) and one sample of calcareous sandstone (BZ-3) display a typical shale-type REY pattern. Carbonate samples display an REY pattern similar to that of modern seawater characterized by LREE depletion (average (Pr/Yb)N = 0.76), slightly negative Ce anomalies (average Ce/Ce* = 0.88), and super-chondrite Y/Ho ratios (average = 45.4). However, unlike the modern seawater REY pattern, 13 of 15 analyzed carbonate samples of the Upper Buzhai reef section display positive Eu anomalies (Eu/Eu* = 0.93–1.94; average = 1.36) and MREE enrichment ((Tb/Yb)N = 1.26–2.42; average = 1.48).

Figure 7 PAAS-normalized REY patterns for Buzhai reef section samples.

Modern seawater and river water REY patterns are from Wang et al. (2018). Refer to Fig. 2 for sample locations.

Five analyzed clastic sedimentary samples are characterized by Th and Zr concentrations (average = 5.89 ppm and 107 ppm, respectively) greater than those of carbonate samples (average = 0.646 ppm and 5.21 ppm, respectively). Among carbonate samples, reefal limestone samples are characterized by less Zr, Th, REY, and greater Y/Ho than are bioclastic packstone samples. Like the Jiwozhai reef section, samples of the Buzhai reef section display positive correlations of Th and Zr and TREY and Th (Figs. 8A and 8B). Total TREY of analyzed carbonate samples ranges from 2.96 ppm to 57.05 ppm (average = 10.89 ppm) compared with an average TREY value of detrital sedimentary samples of 78.80 ppm.

Figure 8 Cross-plots of Y/Ho-(Sm/Nd)PAAS for carbonates in the Devonian.

The mixing lines are calculated with an assumption of conservative mixing and that the Nd concentration of fresh water is 60 fold that of seawater. The Y/Ho of the seawater end-member is assumed to be the same with the highest value of the Buzhai samples (52.9); while the Y/Ho of the fresh water end-member is assumed to be 27.5. The other two inter-REE ratios of the two end-members are near to the results of least square linear fitting, with a small adjustment to fit the data well.

The Upper Buzhai reef section presents a similar δ18Ocarb trend to the Jiwozhai section with the higher values (−4.25 ‰∼−7.10 ‰) in the middle and lower part and lower values (−7.15 ‰ to −16.12 ‰) in the upper part of the section.

Comparison of the Upper Buzhai and Jiwozhai reef sections

The Upper Buzhai section carbonate sample suite is characterized by low immobile elements (e.g., Th and Zr) and TREY concentrations and siliciclastic minerals, elevated Y/Ho ratios, and δ18Ocarb values relative to the Jiwozhai section. Carbonate deposits of the Buzhai section display seawater-like PAAS-normalized REY patterns whereas carbonate samples of the Jiwozhai section are characterized by shale-type PAAS-normalized REY patterns. Reefal limestone samples of the Upper Buzhai and Jiwozhai sections are variably dissimilar to the PAAS-normalized modern seawater REY pattern.

Discussion

Assessment of diagenetic alteration

Given the very high partition coefficients of REY between calcite and seawater (Zhong & Mucci, 1995; Webb & Kamber, 2000; Zhao & Zheng, 2014; Della Porta, Webb & McDonald, 2015), diagenetic models suggest that unrealistically large water–carbonate ratio would be required to reset the REY pattern of carbonate deposits (Banner & Hanson, 1990). The Mn/Sr ratio has been widely used to identify the effects of meteoric diagenesis on primary carbonate. In general, Mn/Sr values > 1 suggest that carbonate has been affected by meteoric diagenesis (Jacobsen & Kaufman, 1999). Mn/Sr ratios of the Upper Buzhai section samples range from 0.27 to 1.15 and those of four carbonate samples (JWZ-6, JWZ-8, JWZ-13, JWZ-14) with Th < 1 ppm in the Jiwozhai section range from 0.23 to 3.76 suggesting that carbonates of the studied sections experienced little meteoric alteration. Two carbonate samples (BZ-4 and BZ-9) of the Upper Buzhai reef section and one carbonate sample (JWZ-14) of the Jiwozhai section with Mn/Sr ratio greater than 1 have the marine PAAS-normalized REY patterns argued that meteoric alteration has the limited effect to REY patterns of the studied carbonate samples. This is also supported by oxygen isotopic values of the carbonate samples from the studied two sections. Two carbonate samples (BZ-15 and BZ-16) from the Upper Buzhai reef section and four carbonate samples (JWZ-4, JWZ-8, JWZ-13, JWZ-14) from the Jiwozhai reef section have the δ18Ocarb values below −10‰, which is conceived to be altered by meteoric diagenesis (Jacobsen & Kaufman, 1999). These samples also share similar PAAS-normalized REY patterns to modern seawaters (Figs. 5 and 7). Moreover, although aragonite is characterized by low partition coefficients of REY compared to calcite, it is likely that REY compositions and patterns are retained during aragonite transformation to calcite (Webb et al., 2009). Studies of modern marine limestones subjected to variable degrees of diagenesis support the survivability of REY distribution patterns in limestone deposits subjected to meteoric processes, marine burial diagenesis, and dolomitization (Webb et al., 2009; Della Porta, Webb & McDonald, 2015; Liu et al., 2019; Luo et al., 2021). Therefore, it is likely that REY compositions and patterns of analyzed carbonate samples of the studied sections were minimally affected by diagenesis.

Evaluation of freshwater contribution

Jiwozhai section

The shale-like REY patterns illustrated by Jiwozhai carbonates deviate from those of modern oxic seawater characterized by HREE enrichment, super-chondrite Y/Ho ratios > 40, and negative Ce anomalies (Fig. 5). In contrast, these REY distributions are similar to those documented from continental or estuarine water characterized by minor Ce anomalies, weak HREE enrichment, variable MREE enrichment, and equal to or slightly greater than the chondrite Y/Ho ratio (Elderfield, Upstill-Goddard & Sholkovitz, 1990; Zhao et al., 2021). Such REY patterns could also have been produced by terrestrial contamination due to the elevated REY contents in shale relative to those of pure carbonate rocks. Indeed, approximately 2% siliciclastic contamination, which is enough to modify the REY composition and pattern of carbonate (Frimmel, 2009; Zhao & Zheng, 2017), corresponds to an upper threshold Th value of 0.28 ppm. Accordingly, carbonate samples containing < 0.28 ppm Th should display REY patterns similar to that of modern seawater. It is noteworthy, however, that the two samples (JWZ-8 and JWZ-13) having the lowest Th contents of 0.194 ppm and 0.059 ppm also display flat REY patterns (Fig. 5) and equal Y/Ho ratios (34.4 and 35.3), neither of which cannot be attributed to silicate contamination. However, the non-marine origin of the analyzed carbonate samples is at odds with the presence of coral, stromatopora, and Brachiopoda (Figs. 4B–4H) as described in ‘Geological Setting’. Therefore, riverine water input to coastal waters appears to have impacted the geochemistry of Jiwozhai section carbonates deposited during the Givetian time. The proportion of freshwater addition to shallow seawater is about 5% as estimated by the cross-plot of (Y/Ho) vs. (Sm/Nd)N for carbonate samples with Th < 1 ppm (Fig. 8).

The Upper Buzhai reef section

REY patterns of most reefal limestone samples of the Upper Buzhai reef section are similar to normal seawater, including LREE depletion, negative Ce anomalies, and elevated Y/Ho ratios (>40). Three bioclastic packstone samples present slightly lower Y/Ho values (35.3 to 39.1; average = 35.6). Although terrestrial contamination cannot be ruled out, a Y/Ho ratio of 40 and Th content of 0.024 ppm of one reefal limestone sample (BZ-12) suggests some degree of freshwater contamination. Moreover, four samples (BZ-9, BZ-11, BZ-12, and BZ-18) contain < 0.1 ppm Th, low (Pr/Tb)N ratios (0.45 to 0.52; average = 0.49) and elevated (Tb/Yb)N ratios ranging from 1.26 to 1.72 (average = 1.47). Low Y/Ho ratios and MREE enrichment displayed by samples characterized by low Th content are attributed to the mixing of riverine water with seawater. The addition proportion of riverine water is below 1% as estimated by cross-plot of (Y/Ho) vs. (Sm/Nd)N for carbonate samples with Th < 1 ppm (Fig. 8), which is much less than that of the Jiwozhai reef section. It is consistent with the paleogeographic location of the two studied reefs and is also supported by the fact that carbonate samples in the Jiwozhai section have lower oxygen isotopic values than those in the Upper Buzhai section.

Terrestrial clastic contamination evaluation

The greater content of REY in shale than carbonate necessitates consideration of the possible role of terrestrial clastic contamination of the studied Jiwozhai Formation carbonate samples. High field strength elements such as Th and Zr are rarely susceptible to chemical weathering and diagenesis (Frimmel, 2009). This supposition is supported by the positive correlation of Th and Zr concentrations of the studied samples (Figs. 6A and 9A). These elements are widely utilized to evaluate the extent of terrestrial sediment contamination of carbonate deposits (Frimmel, 2009; Zhao & Zheng, 2014; Zhao et al., 2021). Elevated contents of REY should be expected in carbonate samples that experienced greater degrees of terrestrial contamination as suggested by the positive correlation of TREY and Th (Figs. 6B and 9B). Terrestrial sediment contamination appears to have affected Jiwozhai carbonate samples as suggested by their shale-like REY patterns (Fig. 5). Therefore, both Jiwozhai and Buzhai reefs appear to have experienced terrestrial input during the Givetian time.

Figure 9 (A) Th vs. Zr cross-plot for Buzhai reef section samples; (B) Th vs. TREY cross-plot for Buzhai reef section samples.

Elevated Th contents (average = 1.97 ppm) and shale-like REY patterns of Jiwozhai section samples compared to those of the Buzhai section suggest that the depositional site of the Jiwozhai section experienced a greater terrestrial input than did the depositional site of the Upper Buzhai section, an argument supported by the paleo-geographic location and fossil assemblages of the Jiwozhai and the Upper Buzhai sections. That is, the Jiwozhai patch reef was located much closer to the Givetian shoreline than was the Buzhai platform margin reef (Fig. 1). As described earlier (‘Geological Setting’), stromatoporoids appear to have been more abundant in the Upper Buzhai reef than in the Jiwozhai reef. Stromatopora is a calcified sponge (Kershaw, 1998) that favors clear seawater that received minimal terrestrial input (Kershaw, 1998; Königshof & Kershaw, 2006). In contrast to stromatoporoid, coral can survive or even flourish in nearshore seawaters as evidenced by the Great Barrier Reef of Australia (Anthony, 1999; Saha et al., 2021).

Ce anomalies

Modern oxic seawater is characterized by significantly negative Ce anomalies in PAAS-normalized REY patterns that reflect the lower solubility of tetravalent Ce than its neighboring La and Pr in seawater (Elderfield, Upstill-Goddard & Sholkovitz, 1990). However, negative Ce anomalies are absent from anoxic waters (Planavsky et al., 2010). Thus, the history of Ce anomalies recorded by carbonate rock successions can be used to trace ocean oxygenation histories (Wallace et al., 2017). Ce anomalies of carbonate samples of the Jiwozhai section average 0.89 and 0.87 in the Upper Buzhai carbonate sample suite, both values markedly greater than the 0.18–0.45 range of modern seawater values (Sholkovitz, Landing & Lewis, 1994). The common presence of coral, stromatopora and brachiopoda fossils in both studied sections excludes the possibility of sampling limestones deposited in a non-marine or anoxic environment. However, the nature of Ce anomalies in samples from both sections can be attributed to freshwater runoff. Terrestrial silicate detritus and freshwater lack Ce anomalies and are characterized by REY contents of one to several orders of magnitude greater than seawater (Tepe & Bau, 2016). Therefore, the introduction of a small amount of terrestrial detritus and freshwater into normal seawater will mask the latter’s original negative Ce anomaly.

Eu anomalies

The Upper Buzhai reef section presents positive Eu anomalies in both reefal limestone and bioclastic packstone samples (average Eu/Eu* of 1.39, n = 15). Positive Eu anomalies (denoted as Eu/Eu*) are commonly cited as evidence of hydrothermal input (Bau, 1991). No visual high-temperature hydrothermal alteration in the Upper Buzhai section in the field and under the polarizing microscope, post-depositional hydrothermal alteration could be excluded. However, enhanced plagioclase weathering induced by greenhouse conditions may also yield positive Eu anomalies (Verdel, Phelps & Welsh, 2018). Moreover, enriched Ba content is known to produce positive Eu anomalies because of Ba interference during ICP-MS analysis though this analytical artifact can be resolved by plotting Eu/Eu* vs. Ba/Eu (Jiang et al., 2007). The latter scenario is excluded as no significant linear correlation exists between Eu/Eu* and Ba/Eu (Fig. 10A). The argument of plagioclase weathering is incompatible with the absence of an Eu anomaly in carbonate samples of the nearshore Jiwozhai reef section (0.91–1.19; average = 1.06). Therefore, the introduction of high-temperature hydrothermal water into seawater was the favored explanation of the positive Eu anomalies displayed by the carbonate sample suite of the Buzhai reefal section. Although Eu2+ in high-temperature hydrothermal fluids is re-oxidized during mixing with ambient cold seawater, the positive Eu anomalies can be recorded in low-temperature precipitates due to the similar geochemical behavior between Eu3+ and its trivalent REY neighbors (Bau et al., 2010; Zhao et al., 2022). This interpretation is buttressed by the occurrence of basalt layers in the Luofu Formation of Guangxi (Liu, Qin & Yan, 2012), and inferred deep-water (Nandan-type) equivalent of the Jiwozhai Formation (Qie et al., 2019).

Figure 10 (A) Eu/Eu* vs. Ba/Eu cross-plot in for Buzhai reef section samples; (B) Eu/Sm vs. Sm/Yb cros-plot for Buzhai reef section samples.

Composition of two end members and the mixing line are from Alexander et al. (2008).

The impact of hydrothermal fluids on ambient seawater can be estimated quantitatively by a simple two-member mixing model (Alexander et al., 2008). One member is modern seawater characterized by low Eu/Sm and Sm/Yb ratios and the other member is high-temperature hydrothermal fluids of much greater Eu/Sm and Sm/Yb ratios. The Sm/Yb versus Eu/Sm cross-plot (Fig. 10B) demonstrates that both ratios of Sm/Yb and Eu/Sm in the Buzhai carbonates can be explained by mixing small (less than 1%) fractions of high-temperature hydrothermal fluid with the seawater. It is likely that high-temperature fluid accounted for less than 1% of the seawater during the accumulation of carbonates in the Upper Buzhai reef section.

Insights into nutrient sources and expansion of reef ecosystem

Coral is sensitive to the input of nutrients and sediment (Schlager, 1981; Hallock & Schlager, 1986). Indeed, the impact of increased nutrient supply on the coral reef ecosystem has become a focus of research in recent years. Upwelling nutrient (e.g., phosphorus and nitrogen)-laden modern deep seawater is known to be an important source of nutrients for some reef ecosystems (Andrews & Gentien, 1982; Eidens et al., 2015; DeCarlo et al., 2021). High-temperature hydrothermal fluids from the basalt altering in the deep water of Dianqiangui Basin were characterized by remarkably positive Eu anomaly, which could be recorded in the chemical precipitates affected by upwelling deep water. However, the presence of weak positive Eu anomalies of carbonate samples of the Upper Buzhai reef section suggests that upwelling was not the dominant source of nutrients for the Buzhai and Jiwozhai reef ecosystems during the Givetian Stage. Indeed, as described above, the elevated content of Th and the deviation of the Jiwozhai Formation carbonate REE patterns from modern seawater points to river runoff being the primary source of nutrients for both the Buzhai and Jiwozhai reef ecosystems. Moreover, the fact that the Jiwozhai reef deposits contain a considerable amount of siliciclastic mineral (e.g., quartz and illite) in the reefal limestone and a greater biodiversity than the Buzhai reef (Liu et al., 2004; Huang et al., 2020) suggests that reef-building was sustained by continental runoff. It is noteworthy that a coral community dominated by tabulate and rugose coral described from the Fanning River area of Queensland, Australia, appears to have thrived in shallow turbid water during Givetian time (Zapalski et al., 2021). It appears that turbid-water reefs were not unusual during the Givetian time.

Widespread platy growth habits of tabulate corals, stromatoporoids, and chaetetids in the Jiwozhai reef biota indicated that they lived in light-limited environments (Huang et al., 2020). The platy morphology of the Givetian reef biota also developed in other parts of Dianqiangui Basin (Liu et al., 2003) and the southern shelf of Laurussia at tropical latitudes (Zapalski et al., 2021). The Jiwozhai reef, together with other Givetian reef ecosystems at tropic latitudes, suggests that Givetian reef biota share functional characteristics with modern scleractinian-dominated turbid-water assemblage. In contrast to the previous view that turbidity and its associated light depletion were regarded to be unsuitable for coral reef development, coral reefs with high biodiversity and coverage have been documented in high turbidity and low light conditions (Richards et al., 2015; Morgan et al., 2016; Mies et al., 2020). Recent studies further proposed that turbid-water coral reefs suffered considerably less bleaching through depleted light penetration under global warming conditions (Sully & Van Woesik, 2020; Mies et al., 2020; Saha et al., 2021). Surface seawater temperature of 30 °C–35 °C was reconstructed by conodont apatite oxygen isotope during the Givetian time from South China and Australia, which means ∼8 °C–10 °C warmings relative to Eifelian (Chen et al., 2021). We propose that tropic corals and stromatoporoids that were adapted to, and possibly even thrived in, turbid waters are responsible for the expansion of the Givetian reefs under higher surface seawater temperatures.

Conclusions

(1) REE geochemistry of carbonate samples of two Devonian reef sections of South China suggest that shale-type PAAS normalized REY patterns of nearshore water in which the Givetian Jiwozhai Formation accumulated differed significantly different from marine water masses on the marginal platform of the Dianqiangui Basin. Together with a considerable amount of siliciclastic minerals and the elevated immobile elements contents in carbonate samples from the Jiwozhai near-shore reef section point to the terrestrial runoff effect during the reef development.

(2) The nutrient source that sustained the reef ecosystem that encompassed the studied Jiwozhai and Buzhai sections appear to have been dominated by terrestrial runoff. Upwelling of nutrient-rich deep water played a minimal role in maintaining the Givetian reef ecosystem. Enhanced terrestrial sediment input associated with Givetian Jiwozhai coral-stromatoporoid reef development demonstrates that the coral ecosystem thrived in turbid waters.

(3) Results of the present study suggest that tropic coral and stromatoporoid adaptation to turbid water played an important role in the Middle Devonian (Givetian) expansion of coral-stromatoporoid reef complexes during the global green-house climate.

Supplemental Information

Supplemental Information 1 Raw data of BZ

Click here for additional data file.

Supplemental Information 2 Raw data of JWZ

Click here for additional data file.

Supplemental Information 3 XRD data of all samples

Click here for additional data file.

We sincerely thank three anonymous reviewers and editors for their constructive comments and helpful suggestions. We are grateful to Professor Yue Wang for the field work help and to Jianxi Long for lithological work under the polarized-light microscope. We are indebted to Dr. Yuanlin Chen for his help for figure drawing and helpful suggestions.

Additional Information and Declarations

Competing Interests

Author Contributions

Data Availability

The authors declare there are no competing interests.

Qi Mao conceived and designed the experiments, performed the experiments, analyzed the data, prepared figures and/or tables, authored or reviewed drafts of the article, and approved the final draft.

Shangyi Gu conceived and designed the experiments, analyzed the data, authored or reviewed drafts of the article, and approved the final draft.

Huan Li conceived and designed the experiments, authored or reviewed drafts of the article, and approved the final draft.

Gary G. Lash conceived and designed the experiments, authored or reviewed drafts of the article, and approved the final draft.

Tianyi Zhang performed the experiments, prepared figures and/or tables, and approved the final draft.

Xiaofeng Xie performed the experiments, prepared figures and/or tables, and approved the final draft.

Zidong Guo performed the experiments, prepared figures and/or tables, and approved the final draft.

The following information was supplied regarding data availability:

The raw measurements are available in the Supplementary Files.

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
