# Peer review of "Rare earth element geochemistry of Middle Devonian reefal limestones of the Dianqiangui Basin, South China: implications for nutrient sources and expansion of the reef ecosystem"

_PeerJ, doi:10.7717/peerj.13663_

## Round 0.1 · original submission · Major Revisions

We have obtained three very detailed and well-thought-out reports. You will see that all reviewers consider that your paper falls quite short of what is required, but they have provided very significant suggestions for its improvement. Your paper may become publishable if you thoroughly address all their questions and if you bring it in line with their expert commentary. Please note that, besides their comments, two of the reviewers have helpfully provided annotated PDFs which you should also consider in your response.

Reviewer 1 ·

Basic reporting

The authors reported some trace elements and REY data from the Middle Devonian reef limestones in South China. The reefal rocks lack significant Ce anomaly and have high REE contents – that’s a strong signal of siliciclastic contamination and diagenesis. However, the authors interpret that these REY values record seawater signature and further they infer that the nutrient supply from rivers was the major driver for reef proliferation during the Givetian time. One of the problems is that the laboratory procedure is not clearly layered out – including the concentrations of acid, the purity of carbonates, the reaction time and data calibration process. The careless analytical procedure lead to the confusion about the diagenesis and siliciclastic contamination. Apparently, the lack of Ce anomaly and middle REE enrichment in reef limestones suggest that the REYs are not recording the ancient seawater signal. Instead, they record either siliciclastic contamination or diagenesis. The siliciclastic contamination is commonly avoided by using very weak acid and by leaching process (Zhang et al., 2015; see below). The diagenetic incorporation of siliciclastically-sourced REYs is difficult to identify but commonly involves very careful petrographic observation and screening. The authors used Th concentration for screening the siliciclastic contamination but at the same time they used Th concentration as a proxy for increased nutrient supply. The major question becomes: What’s the use of these REY data? One can measure a section and qualitatively derive a similar conclusion by looking at the overall lithology of the two sections.

For this paper to be published, the authors should provide a detailed analytical procedure for the trace element and REY analyses, clearly distinguish terrestrial vs. diagenetic (at least petrographic screening) effects on the data, and point out the uncertainties of using REYs in carbonates for paleoenvironmental interpretations (e.g., Ce anomalies are commonly used as a redox proxy for depositional environments; however, in this case they are not recording the redox condition of seawater).

Experimental design

One of the problems is that the laboratory procedure is not clearly layered out – including the concentrations of acid, the purity of carbonates, the reaction time and data calibration process. The careless analytical procedure lead to the confusion about the diagenesis and siliciclastic contamination. Apparently, the lack of Ce anomaly and middle REE enrichment in reef limestones suggest that the REYs are not recording the ancient seawater signal. Instead, they record either siliciclastic contamination or diagenesis. The siliciclastic contamination is commonly avoided by using very weak acid and by leaching process (Zhang et al., 2015; see below). The diagenetic incorporation of siliciclastically-sourced REYs is difficult to identify but commonly involves very careful petrographic observation and screening.

Validity of the findings

The REY data may not record seawater signature - this needs to be justified!

Additional comments

More specific comments are provided below:
Line 29: “the positive correlation of REY and Th contents”: Do not understand the meaning of this. Positive correlation of Th and total REE?
Line 30: “a terrestrial siliciclastic contribution”: Isn’t this a diagenetic feature? How was the Th incorporated into carbonates? Here it needs to be clarified if the terrestrial contribution is caused by the analytical procedure (strong acid) or diagenetic incorporation of Th and other elements into neomorphic carbonate minerals.
Line 35-36: Average Eu/Eu* = 1.35; how can this be negligible? Slightly influenced?
Line 37: again, “positive correlation of REY and Th”: Are you correlating total REE and Th? Remember Y is commonly not included in total REE;
Line 38: “riverine input”: So both sections have a positive REE-Th correlation. The Jiwozhai section is interpreted as terrestrial contribution; the Upper Buzhai reef section is interpreted a riverine input? Aren’t those the same? In addition, why does the marginal platform section have stronger signal for riverine input?
Line 40: “turbid water containing abundant terrestrial sediment”: Shouldn’t this result in deposition of siliciclastic sediments observable in the measured section?
Line 50: “distances of 400 km to 3100 km”: This is ambiguous. Isn’t latitude here more appropriate?
Line 71: REY3+? Not all the REYs have the same attributes, particularly for La and Ce;
Line 78-79: Northdurft et al. (2004) provided an example for the potential use of REEs, but it does not mean REYs from all carbonates can be used as seawater proxy! See reviews in Zhao et al. (2021) [Zhao et al., 2021, Rare earth element geochemistry of carbonates as a proxy for deep-time environmental reconstruction. Palaeogeography, Palaeoclimatology, Palaeoecology, 110443.] and Zhao et al. (2022) [Zhao et al., 2022, A review of retrieving pristine rare earth element signatures from carbonates. Palaeogeography, Palaeoclimatology, Palaeoecology 586, 110765.], in addition to Tostevin et al. (2016) that the authors have cited.
Line 132-139: This paragraph should be combined with the paragraph starting at line 108!
Line 152: The concentration of HNO3?
Line 140-159: Methods: Please be a little more specific about the acid concentration and calibration procedure. The acid used for REE analyses of carbonates is very important for the results. Most researchers would use the very weak acid such as 5% acetic acid or 2% HNO3 (see Zhang et al., 2015, A refined dissolution method for rare earth element studies of bulk carbonate rocks. Chemical Geology 412, 82-91.);
Line 168-169: Because La can easily be modified by source rocks and local processes other than marine, La is commonly not used for calculating the Ce anomaly. See: Lawrence et al., 2006, Rare earth element and yttrium variability in South East Queensland waterways. Aquatic Geochemistry 12, 39-72.
Line 216-232, section 5.1: Apparently the total REEs and other indicators suggested that these rocks are severely influenced by clastic contamination, either caused by the analytic method (strong acid that dissolves siliciclastic components) or by incorporation of REE signal of clastic components (e.g., clays) into carbonates during diagenesis. The authors cannot eliminate any of these possibilities in the paper – it makes the claim “studied sections were minimally affected by diagenesis” weak and the remaining interpretation problematic;
Line 256-258: What’s the difference between terrestrial contamination and freshwater contamination? Why does the Th contents of 0.024 (which is very low) imply freshwater contamination?
Line 306, section 5.5: In this section, the authors need to clarify that the high Eu/Eu* values were not caused by post-depositional hydrothermal fluids! In addition, Eu anomalies from high-temperature hydrothermal fluids cannot be carried to the shelf margin if seawater was oxic: Fe-Mn oxide/crust would sequester REYs before they reach the shelf margin!
Line 354-360: Sr isotopes suggest no enhanced weathering, but the authors suggest enhanced nutrient supply from runoff or riverine input. What specific implication the data suggest?
Fig. 1b: “Dianqiangui Basin” should not cross the line. Adjust the size and fit in the tan areas.
Fig. 2: The stratigraphic correlation violates the basic sedimentological concept. Unless there is unconformities or significant hiatus in the Upper Buzhai section near the top of the reef intervals, there is no accommodation space for the Jiwozhai section to deposit strata 20 times thicker than those of the shelf margin (see the uppermost correlation line).

Reviewer 2 ·

Basic reporting

This article is easy enough to follow, with the proper format, and the figures are fine (I made a suggestion about the REE plots to the authors in the PDF copy of the review). I do think that the authors need to include additional information about the geologic setting (particularly in regards to the basalts) and update their ideas about Ce anomalies and hydrothermal fluids. There is a lot more nuance to those topics than the article provides, and they are oversimplying how these things work and making assumptions based on this oversimplification.

Experimental design

This is where the article falls short; the interpretations are a good first pass at making a hypothesis, but there just isn't enough analysis to support the conclusions. Everything is based on simple geochemistry analyses without any information provided about the non-carbonate fraction's mineralogy, trace element complexation, stable isotope signatures, or hydrothermal plume behavior.

The data as they are make a good jumping off point, but they are just the beginning and you can't make a coherent explanation to fit the data without also analyzing (or considering) the other options. This feels like the beginning of a project... it's interesting preliminary data and the method to get it is fine, but these data do not tell the story that the authors are trying to tell. There is too much missing context.

Validity of the findings

I think the conclusions are based on too little data and too many assumptions. The data provided are reasonable but there is just not enough of it to say the things the authors are saying. This reads like a progress report for the first half of a project, or a thesis prospectus... there is promise in the data but you can't say much about it yet.

Additional comments

I am choosing major revisions rather than reject, because I think that what the authors have here is a good place to *start* a testable hypothesis. But as it stands there are too many assumptions in the paper to accept it as is right now. The study needs considerably more analyses (stable isotopes of oxygen, mineralogy analysis, microtextural analysis, EDS mapping if possible, clay identification for the siliciclastic-rich section, etc. in order to be robust in its assertions. Right now it's all hand-wavy and guess-laden.

Annotated reviews are not available for download in order to protect the identity of reviewers who chose to remain anonymous.

·

Basic reporting

The manuscript by Mao et al. provides trace element geochemical data to test local water quality parameters in onshore and offshore reefs in the Middle Devonian of South China. The use of trace element geochemistry in ancient carbonates has become a very important tool for understanding both local geography and broader seawater chemistry through time. It is a suitable topic for the journal and will find broad interest in Earth historians. Better understanding of the sources for nutrients to a major reef track is an interesting study.
The paper is well written but needs a bit more English editing. I have provided comments on the PDF to help, but be sure the native English reader has a go at the grammar.
That said, the paper could have much better analysis of the trace element data to better understand how different the two settings were. As it stands, the paper does not add much knowledge to the two reef areas. Analogies to modern corals are also a problem, as they are not the same as the ancient ones. Finally, the paper does not really add much to why reefs expanded in the Givetian.
The data files open OK, but probably should be reduced to reliable digits.

Experimental design

I think we need a more complete description of the actual reefs, even if taken from previous papers. Terms like ‘bioclastic limestone’ are very broad and do not convey adequate environmental information for the settings of the reefs. What about local bathymetry? It is not mentioned how laterally extensive the reefs are or if they have lateral zonations/variability (as all reefs do). It would be good to get an idea of exactly what facies are present.
You need to clearly state exactly what part of the limestones was sampled!! The difference between particulate micrite and a radiaxial cement is huge and does not necessarily provide much information about the water itself. (see Della Porta et al 2015, which you cite) Interpreting the data is very difficult and misses some key possibilities when it is not know what was actually sampled. (Even the sandstone sample is just labelled sandstone!! – what kind of sandstone?)
It is very important to use the equations of Lawrence et al., 2006 (ref on PDF) for REY anomaly calculations as the equations being used use elements that themselves display anomalous behaviour in seawater to calculate the anomalies of other elements. La and to a lesser degree Gd have seawater anomalies.
Figs 5 and 7 really need to have the sample types shown with different patterns/colours. Make it possible to identify which are ‘reefal limestone’ and which are other sed types at a glance. It is not good to have to look back at the strat columns. Then, you really need to plot the different sed types on X-Y plots showing LREE depletion, Y/Ho , and other anomalies to see how different they really are. Then you can see trend lines from the more pure limestone to the more contaminated limestone. Some of the Jiwozhai samples (e.g., 6) have decent sea water like patterns. You cannot just say they are all flat. That would provide a means of really determining what was likely form the water and what was form direct siliciclastic contamination in the sample. You could then compare those data to the other trace elements and possibly even do mixing lines to determine how much contamination existed. However, as it is not clear what a reefal limestone sample really is (all Stromatoporoid, or coral or mud between allochems), it is still a problem.
The diagenetic vetting does not seem to have been done very well and at least needs to be acknowledged in the results.

Validity of the findings

Although the basic findings of the study that the Jiwozhai section is more affected by siliciclastic detritus than the upper Buzhai section is clearly supported, the evide3nce form REE has not been used very effectively, and does not seem to provide much advance on what is known form the petrography in the first place. The data may be able to be used with greater detail, but it is not clear what was actually analysed in each sample and the lack of cross plots makes it hard to compare them. I think there could be more here than has been presented, but as is, the paper does not make much of a contribution. The three basic conclusions are: one reef is more inshore compared to the other (which was already known), there was terrestrial runoff – (which was already known form the petrography), and the third conclusion that corals and stromatoporoids in turbid environments played an important role in the expansion – does not have any evidence – as it was also turbid before the Givetian!!
The interpretation of bathymetry based on thickness of limestone is not supportable. Thickness may be limited by bathymetry at a given time, but any thickness can accumulate in any bathymetry through time.

Additional comments

I have provided some English editing on the PDF, but it is mostly well written.
Fig. 1 – on figure ‘Inter-plaiform basin’ should be ‘Inter-platform basin’ As the palaeogeography is critical here, it would be nice to see more of a palaeogeography on part c as well – to show where the platform edge is and the coast line.
A new figure showing a palaeogeographic reconstruction of the two sections would be very useful, even if at a very schematic level. Paleogeography is a critical point of the study but is not shown very well or in any detail.
The paper needs additional figures to relate the geochemistry of the two regions better (as mentioned in the PDF and above.

---

## Round 0.2 · Minor Revisions

You will see that reviewer#2 is quite satisfied with your changes, although reviewer #1 still has some misgivings regarding the possible seawater vs. clastic contaminations. It may be impossible to completely allay reviewer#1's concerns, but I am willing to accept your paper subject to addressing the other remaining minor issues highlighted by our reviewers, and inclusion of some discussion regarding the possibility of such contamination.

Reviewer 1 ·

Basic reporting

I have read through the responses to previous review comments and the manuscript. The authors have tried their best but I do have concerns if the data can tell a clear story. However, asking the authors for another revision is perhaps not going to make a difference. There are some specific comments made below:
Line 41-44: Turbid water provided refuge for reef builders: This is a very weird and irrational conclusion – Maybe ‘the nutrient-laden water helped expand reef-builder habitats…”
Line 88-90: This is weird – yes, pristine REE can be recovered but it has very strict criteria to recover. You need to compare with their criteria to your study, not just citing these two papers saying it’s recoverable. There are many papers earlier than these two that have made the same claim!
Line 172-175: Are not carbon and oxygen isotopes measured at the same time?
Line 199-202: Again, wasn’t carbon isotopes also measured?
Line 211-212: Linear scale: In Lawrence et al. (2006), they clearly stated that the geometric calculation (equation 7 in their paper) is the better choice. Why do the authors choose the linear calculation?
Line 203, section 4, results – why don’t you at least plot the data against the stratal columns of these two sections?
Line 337, section 5.3.: This section becomes a serious problem. Terrestrial components have REE concentrations orders of magnitudes higher than seawater and carbonates. If you consider that these REE data have already influenced by terrestrial contamination, then the meaning of the REY data for seawater interpretation is problematic. My major concern for this dataset is that it was produced using the HNO3-HF (1: 4) acid and it is difficult to distinguish which part was from the carbonate portion and which part represents the province of clastic input.
Line 407, 5.6: what aspects of the REE data (the major analyses of this ms) help to strength the argument in this section? Looks like the conclusion is mainly from the Th concentration or a hypothesis that does not the data from this manuscript.

Experimental design

The major concern is the whole-rock REE analyses - to what degree the REY data record seawater signature or a mixed signature of clastic contamination is not clear. Do the isotope analyses also include carbon isotopes? The authors only report the oxygen isotopes.

Validity of the findings

Can focus on the nutrient supply - the argument that turbid water promote proliferation of reef builder seems to be problematic.

Additional comments

I don't know what suggestions I can provide to the authors - my concern is that the whole rock analyses may not record the seawater signature and it is difficult to distinguish seawater vs. clastic contamination in the geochemical results.

Reviewer 2 ·

Basic reporting

I have a few minor edits for language (on the attached PDF), but overall this looks good. The lit review has been expanded. The figures are MUCH better.

Experimental design

This revision is a much better paper than its original submission. The oxygen isotope data and XRD/mineralogy discussion adds a lot of necessary background and support to their interpretations, as suggested by multiple reviewers.

Validity of the findings

These all looked fine.

Additional comments

I'm selecting minor revisions due to some typos, but these are really very minor (basically can be done in a few minutes). Overall, the authors should be commended for their revisions, and as a reviewer I thank them for taking the suggested changes seriously and following through with them.

This paper is MUCH better than the original submission, and tells a coherent story that is now backed up by sufficient data.

Annotated reviews are not available for download in order to protect the identity of reviewers who chose to remain anonymous.

---

## Round 0.3 · accepted · Accept

Thank you for addressing the last few issues!